# The impact of malaria-protective red blood cell polymorphisms on parasite biomass in children with severe *Plasmodium falciparum* malaria

S. Uyoga [1,6✉], J. A. Watson [2,3,6✉], P. Wanjiku[1], J. C. Rop[1], J. Makale[1], A. W. Macharia [1], S. N. Kariuki[1], G. M. Nyutu [1], M. Shebe[1], M. Mosobo[1], N. Mturi[1], K. A. Rockett[4], C. J. Woodrow[2,3], A. M. Dondorp [2,3], K. Maitland [1,5], N. J. White [2,3] & T. N. Williams [1,5✉]

Severe falciparum malaria is a major cause of preventable child mortality in sub-Saharan Africa. Plasma concentrations of *P. falciparum* Histidine-Rich Protein 2 (*Pf*HRP2) have diagnostic and prognostic value in severe malaria. We investigate the potential use of plasma *Pf*HRP2 and the sequestration index (the ratio of *Pf*HRP2 to parasite density) as quantitative traits for case-only genetic association studies of severe malaria. Data from 2198 Kenyan children diagnosed with severe malaria, genotyped for 14 major candidate genes, show that polymorphisms in four major red cell genes that lead to hemoglobin S, O blood group, α-thalassemia, and the Dantu blood group, are associated with substantially lower admission plasma *Pf*HRP2 concentrations, consistent with protective effects against extensive parasitized erythrocyte sequestration. In contrast the known protective *ATP2B4* polymorphism is associated with higher plasma *Pf*HRP2 concentrations, lower parasite densities and a higher sequestration index. We provide testable hypotheses for the mechanism of protection of *ATP2B4*.

[1] KEMRI-Wellcome Trust Research Programme, Kilifi, Kenya. [2] Mahidol-Oxford Research Unit, Faculty of Tropical Medicine, Mahidol University, Bangkok, Thailand. [3] Centre for Tropical Medicine and Global Health, Nuffield Department of Medicine, University of Oxford, Oxford, UK. [4] Wellcome Trust Centre for Human Genetics, University of Oxford, Oxford, UK. [5] Department of Surgery and Cancer, Institute of Global Health Innovation, Imperial College, London, UK. [6] These authors contributed equally: S. Uyoga, J. A. Watson. ✉email: suyoga@kemri-wellcome.org; jwatowatson@gmail.com; twilliams@kemri-wellcome.org

*P*lasmodium falciparum malaria has been the dominant cause of child mortality in tropical regions of Africa and Asia for much of the last 5000 years[1]. As a consequence, malaria has played a major part in shaping the human genome through the selection of mutations that confer a survival advantage[1]. An estimated one-quarter of the variability in the overall risk of malaria is heritable, a figure that reaches one-third for severe and complicated infections[2].

Numerous candidate malaria protective polymorphisms have been proposed during the 70 years since, through his "malaria hypothesis", JBS Haldane first suggested that the high prevalence of β-thalassemia in the Mediterranean basin might be reflective of natural selection by malaria[3]. Although many of these candidates have not held up to detailed scrutiny[4] there is now strong evidence in support of some. The most convincing relate to red blood cells[5] and are particularly plausible given the central role of the erythrocyte in the biology of malaria[6].

Through a recently conducted case-control study, we investigated associations between mutations in a wide range of candidate genes and the risk of severe *P. falciparum* malaria in Kilifi, Kenya. We found significant evidence in support of polymorphisms in 15 genes: seven (*ABO, ATP2B4, INPP4B, G6PD, HBA, HBB,* and *FREM3/GYP*) involved in red cell structure or function, seven (*ADGRL2, CAND1, RPS6KL1, GNAS, ARL14, CD40LG, IL10*) non-red blood cell protein coding genes, two of which related to immune pathways, and one non-coding RNA gene (*LOC727982*)[7]. Although all were associated with a lower risk of severe malaria, only three were associated with lower peripheral parasitemia. Parasite densities were approximately fivefold lower in both heterozygotes and homozygotes for the rs334 A>T allele in *HBB*, the βˢ mutation which encodes for the structural hemoglobin (Hb) variant HbS, in comparison to wild-type normal children, while they were marginally lower in both children with blood group O and in homozygotes for the derived allele at rs1541255 A>G in *ATP2B4*.

However, peripheral parasite densities are a poor measure of total parasite loads in *P. falciparum* infections because of the time-varying proportion of parasites that are sequestered in the deep microvasculature. Although these sequestered parasitized erythrocytes are central to the pathophysiology of severe and complicated malaria, they are not reflected in the peripheral parasite count[8]. Total parasite biomass, which includes both the circulating and sequestered fractions, is better reflected by the plasma concentration of *P. falciparum* Histidine-Rich Protein 2 (*Pf*HRP2)[9], a 30 kD molecule released from the cytoplasm of falciparum-infected red blood cells, mainly during schizont rupture[10–12]. Plasma levels of *Pf*HRP2 have diagnostic utility in differentiating between "true" severe malaria and other severe illnesses with incidental parasitaemia[13].

In this work we show that plasma *Pf*HRP2 and the ratio of plasma *Pf*HRP2 to the circulating parasite density are powerful quantitative phenotypic traits for case-only genetic association studies in severe malaria. We investigate associations between the malaria-protective polymorphisms identified in our earlier study[7] and total parasite biomass estimated through plasma concentrations of *Pf*HRP2, in a cohort of children admitted to hospital with a diagnosis of severe *P. falciparum* malaria in Kenya. The associations with parasite biomass help understand the mechanisms of protection for the studied polymorphisms.

## Results

After multiple imputation, we had data on a total of 2198 patients for our final analyses. The pattern of the missing data is shown in Supplementary Fig. 1. The key predictors of the plasma *Pf*HRP2 concentration were the admission platelet count (log transformed

as previously described)[13], the admission hemoglobin concentration, and parasite density. The overall distributions of parasite densities, plasma *Pf*HRP2 concentrations, and ratio of *Pf*HRP2 to parasites are summarized in Fig. 1a–c while the results of the linear quantitative trait association models between each of these parameters and the 14 severe malaria associated genetic polymorphisms are shown in Fig. 1d. We also performed sensitivity analyses with covariate adjustment, and using the complete case data only (Supplementary Figs. 2–4).

As reported previously, only three polymorphisms were associated with lower parasite densities in cases (Fig. 1d: *HBB*: reduction in geometric mean of 70% [95% CI 50–80]; *ATP2B4*: reduction of 30% [95% CI: 0–51]; *ABO*: reduction of 22% [95% CI: 5–36]). In comparison, four of the six major red blood cell polymorphisms (*HBB, HBA1-2, FREM3/GYP, ABO*) were associated with lower plasma *Pf*HRP2 concentrations, while each had estimated larger effect sizes (Fig. 1d: *HBB*: reduction in geometric mean of 83% [95% CI: 73–89]; *ABO*: reduction of 43% [95% CI: 31–53]; *FREM3/GYP*: reduction of 41% [95% CI: 18–57]; *HBA*: reduction of 37% [95% CI: 15–53]). For these four polymorphisms we compared the best fitting models of association in this case-only analysis to the best fitting models of association from previously reported case-control analyses. There were notable differences with previously published case–control studies regarding the best fitting models for these major polymorphisms. For rs334 (HbS) a heterozygous model of association was reported in the largest case–control study conducted to date, which included almost 12,000 cases and more than 17,000 controls, showing no apparent protective effect of homozygous sickle cell anemia (HbSS)[5]. However, in this case-only analysis of *Pf*HRP2 as a quantitative trait, the best fitting model of association was additive (i.e. with greater protection in homozygotes than in heterozygotes). This offered strong evidence for rejecting the heterozygous model (likelihood ratio test comparing the heterozygous model with the two-parameter model: $p = 10^{-5}$). For *HBA1-2* (α$^{-3.7}$-thalassemia) the best fitting model was recessive, although the data were also consistent with an additive effect (only the heterozygous model was rejected, $p = 0.001$). The best fitting models for the rs186873296 mutation in *FREM3* (Dantu), and the O blood group mutation in *ABO* (rs8176719) assumed additive and recessive models of association respectively, consistent with our previous study[7]. Polymorphisms in *ATP2B4, G6PD, RPS6KL1, LOC727982, ARL14, LPHN2, IL10, CAND1,* and *GNAS* were not associated with differences in baseline *Pf*HRP2 and so no best model of association could be determined.

For two key polymorphisms, the α$^{-3.7}$ deletion in *HBA* and a SNP in *FREM3* which tags the Dantu blood group, no significant associations were seen for parasite density but significant associations were seen for the plasma *Pf*HRP2 concentration ($p = 0.007$ and $p = 0.002$, respectively; Fig. 2). Similarly, while only a marginal association was seen with parasite density for *ABO* ($p = 0.01$), a highly significant association was seen with plasma *Pf*HRP2 concentration ($p = 10^{-7}$, Fig. 2). A notable outlier was the rs1541255 polymorphism in *ATP2B4* (Fig. 3). This polymorphism was associated with the third strongest level of protection in our previous multi-country case-control analysis[14] and was associated with marginally lower peripheral parasite densities ($p = 0.05$). However, in the current analysis we found a trend towards higher plasma concentrations of *Pf*HRP2 ($p = 0.08$), an effect in the opposite direction to that observed for the other malaria protective erythrocyte genetic polymorphisms. As a result, the association with the sequestration index (the *Pf*HRP2/parasite ratio) was significant (70% increase in the geometric mean ratio [95% CI: 20–140], $p = 0.001$). Hemoglobin concentrations on admission were no lower in homozygotes than they were in wild type or heterozygotes, as would be expected if

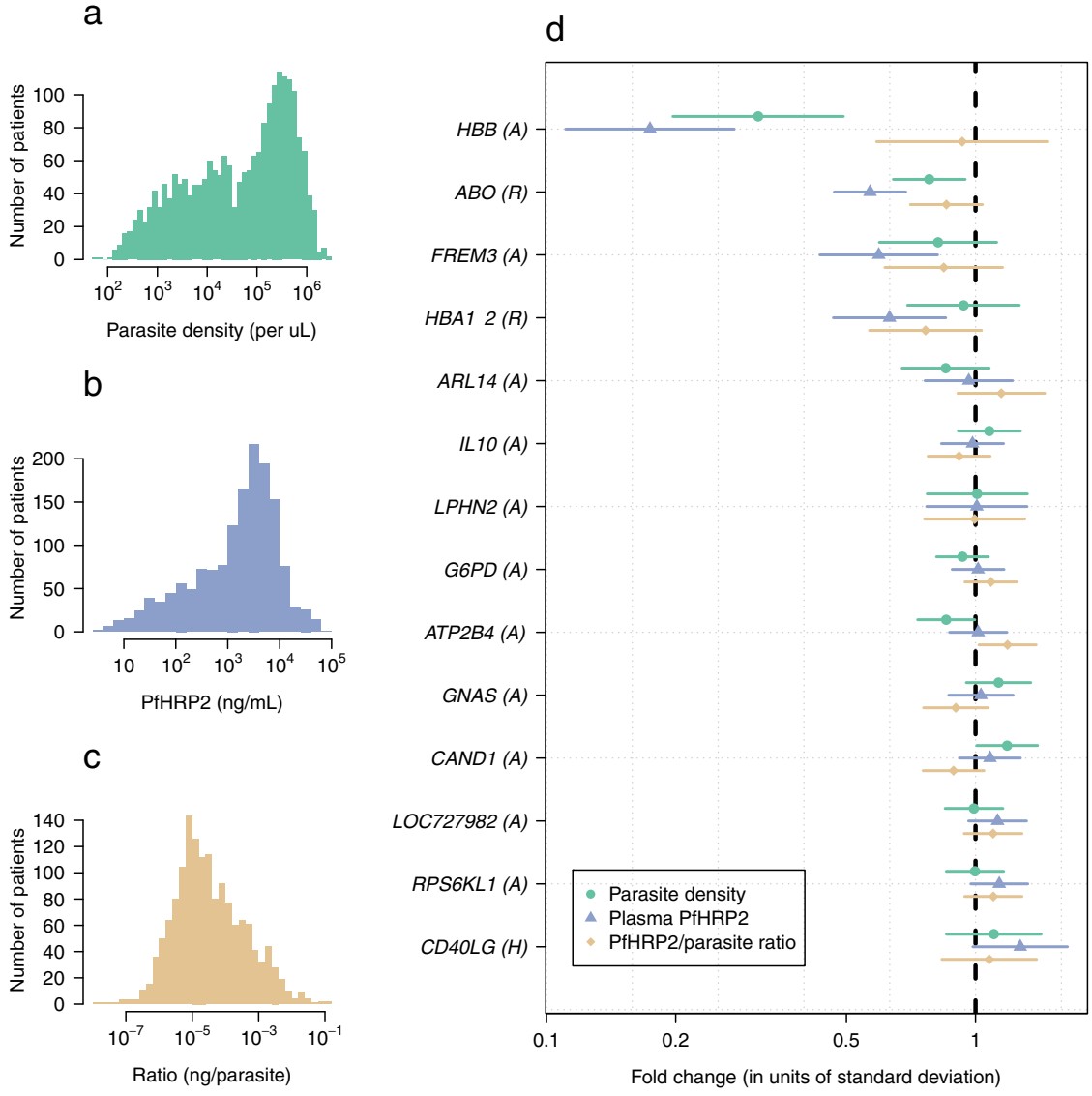

**Fig. 1 Case-only genetic association study using 14 targeted variants and three quantitative traits. a–c** Distributions of the three quantitative traits (observed data: parasite density in green; *Pf*HRP2 in purple; ratio *Pf*HRP2/parasites in orange). **d** Effect estimates (95% confidence intervals) expressed as fold changes on the standardized trait distribution (rescaled to have mean 0 and standard deviation 1).

the higher plasma *Pf*HRP2 concentrations were a consequence of a longer duration of illness (Supplementary Fig. 5).

We re-estimated case-control odds-ratios under a data-tilting framework for the five major red blood cell genetic polymorphisms evaluated using previously published probability weights based on the plasma *Pf*HRP2 and the platelet count to downweigh probabilistically data from patients who were unlikely to have true severe malaria (Fig. 4)[13,15]. This increased the strength of the evidence that *HBB*, *ABO*, *HBA1-2*, and *FREM3/GYP* are associated with protection against severe malaria compared with the non-weighted case–controls odds-ratios (for *HBA1-2*: $p = 10^{-5}$ versus $p = 10^{-4}$; *ABO*: $p = 10^{-13}$ versus $p = 10^{-8}$; *FREM3*: $p = 10^{-12}$ versus $p = 10^{-11}$). In contrast it decreased the evidence ($p = 0.02$ versus $p = 0.003$) for *ATP2B4*.

## Discussion

In this large cohort of children with severe *P. falciparum* malaria, we show that the plasma *Pf*HRP2 concentration is a useful phenotypic measure that can be used to validate genetic associations discovered through case–control studies. Consistent with

previous reports, plasma *Pf*HRP2 concentration is a better measure of disease severity than peripheral parasite density, and more accurately identifies true severe malaria. Our study shows that by providing a simple quantitative assessment of both disease severity and diagnostic accuracy, plasma *Pf*HRP2 concentration can be a useful quantitative trait in genetic association studies.

During the seven decades since the "malaria hypothesis" was first proposed[3] epidemiological support has accumulated for a number of candidate genes. However, in most cases the underlying mechanisms remain incompletely understood[4]. For example, several different hypotheses have been put forward for the best-established candidate, the sickle mutation in *HBB*. The mechanisms proposed fall into three broad categories. First, the early in vitro studies suggested that HbS containing red blood cells are less supportive of *P. falciparum* parasite growth and development, particularly under conditions of low oxygen tension[16–18]. Second, there is evidence that HbS affects the cytoadhesion of late-stage parasite-infected red cells to the vascular endothelium through reduced intracellular transport and expression of surface proteins, thereby attenuating the

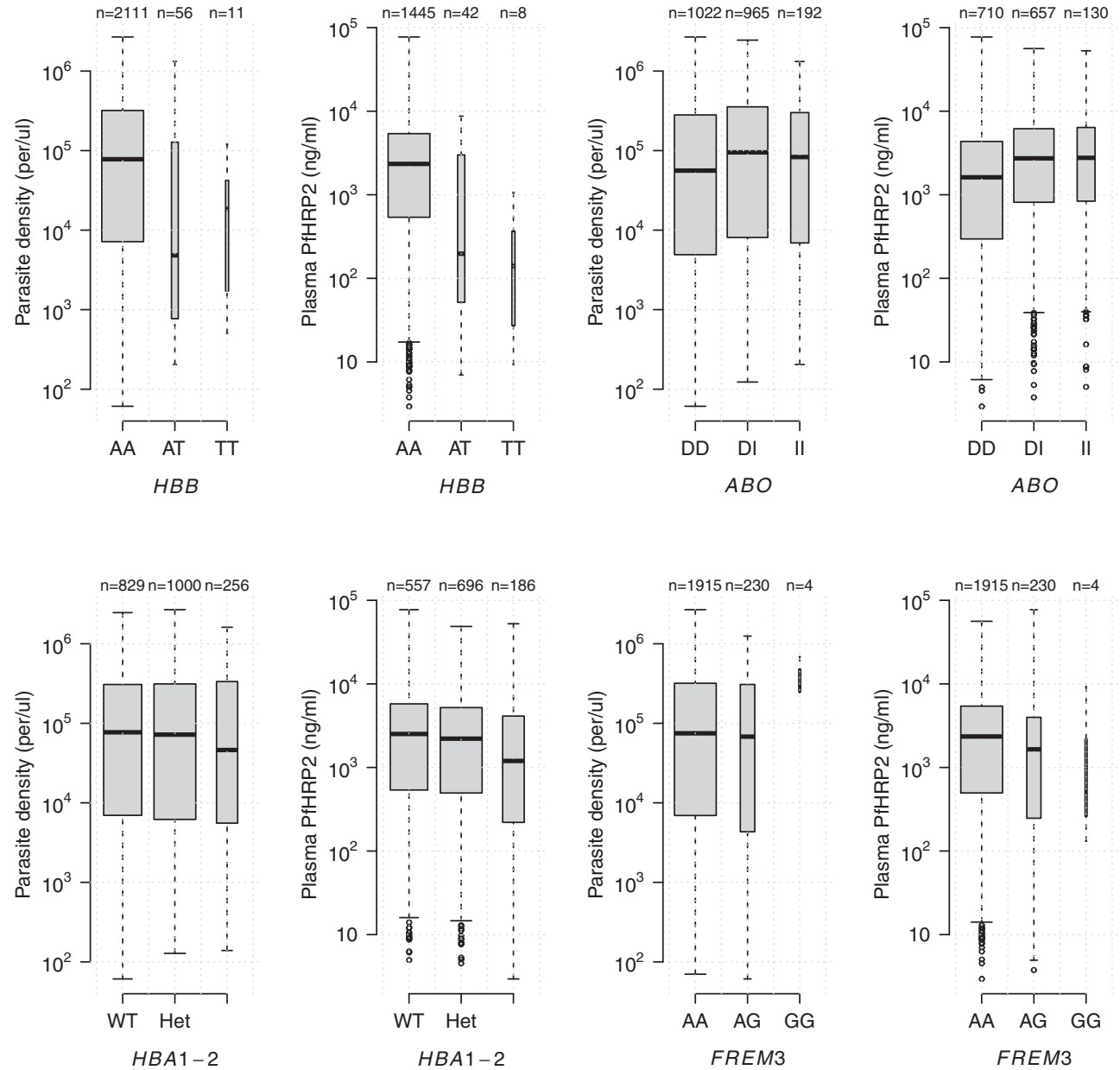

**Fig. 2 Distributions of plasma *Pf*HRP2 and parasite density across *HBB* (rs344), *ABO* (rs8176719), *HBA1-2* (α-thalassemia), and *FREM3/GYP* (rs186873296) genotypes.** Boxplots (center: median; bounds of box: interquartile range; whiskers: 1.5 standard deviations; points: outliers) are based on observed data; the width of each boxplot is proportional to the square root of the number of data points.

pathophysiological consequences of malaria[19–21]. Finally, it has been suggested that malaria parasite-infected HbS containing red blood cells may be removed more rapidly from the circulation through immunological mechanisms, particularly involving the spleen[22–24]. Our observation that both peripheral parasite densities and plasma *Pf*HRP2 are considerably lower in children carrying the β^s mutation is broadly compatible with all these three mechanisms. Similar mechanisms have been proposed for the malaria-protective effect of α-thalassemia yet, as in our current study, no associations have been observed with peripheral parasite densities in the majority of epidemiological studies[4]. Our novel finding that plasma PfHRP2 concentrations are substantially lower in children with α-thalassemia supports the conclusion that *P. falciparum* parasites are less able to thrive in carriers of this condition than they are in normal children. Finally, in a recent mechanistic study we found that red cells

expressing the rare Dantu blood group are resistant to invasion by *P. falciparum* merozoites because they are less deformable than normal cells[25]. While in the current study this was not reflected by reduced parasite densities, we observed a 40% reduction in *Pf*HRP2 concentration in Dantu children overall. Furthermore, the in vitro data showed a greater impact of Dantu for homozygous relative to heterozygous cells[25], consistent with our in vivo data which suggest an additive association with plasma *Pf*HRP2.

An unexpected finding of our study was the relationship between the polymorphism in *ATP2B4* and the plasma *Pf*HRP2 concentration and the sequestration index. The polymorphism involved, which results in red cell dehydration and raised concentrations of intracellular calcium[26], was the third strongest protective signal in the largest multi-country case–control study of severe malaria conducted to date[14]. Nevertheless, in the current study we found that although the protective allele was associated

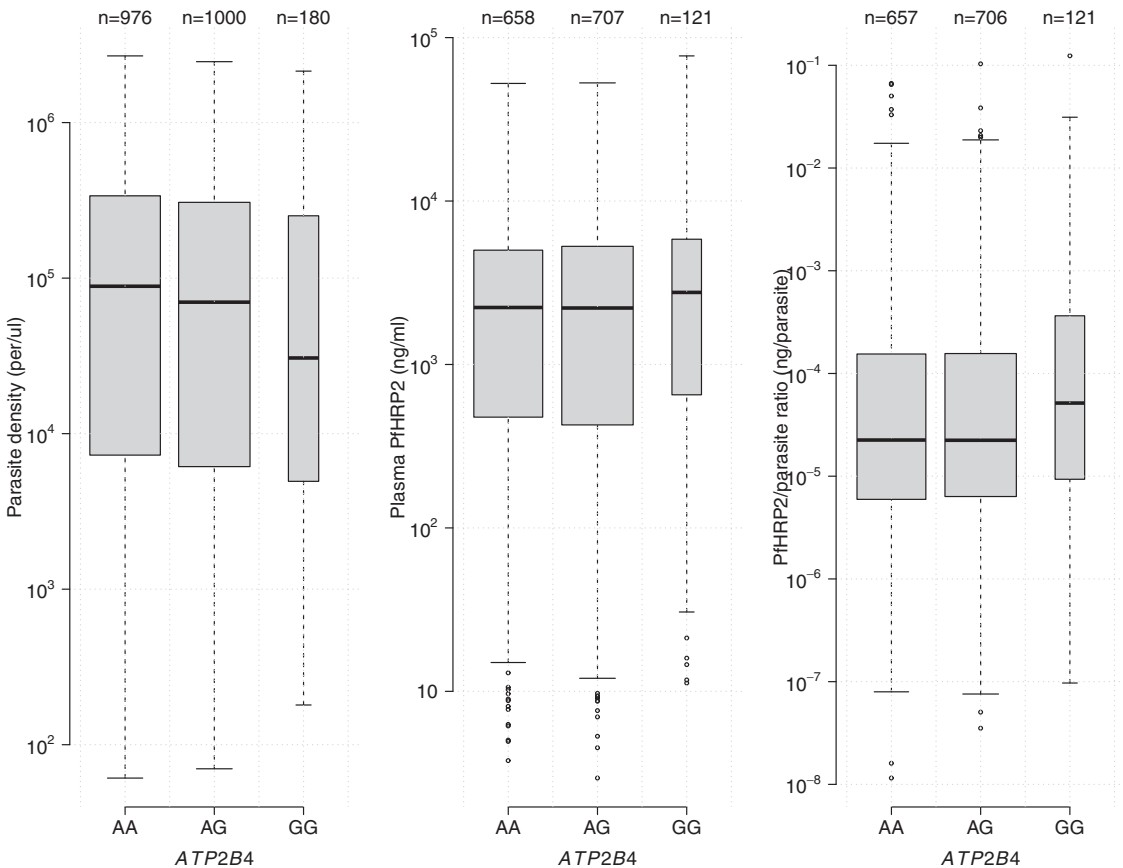

**Fig. 3 Distributions of plasma *Pf*HRP2, parasite density, and *Pf*HRP2/parasite ratio across the *ATP2B4* genotypes (rs1541255).** Boxplots (center: median; bounds of box: interquartile range; whiskers: 1.5 standard deviations; points: outliers) are based on the observed data; the width of each boxplot is proportional to the square root of the number of data points.

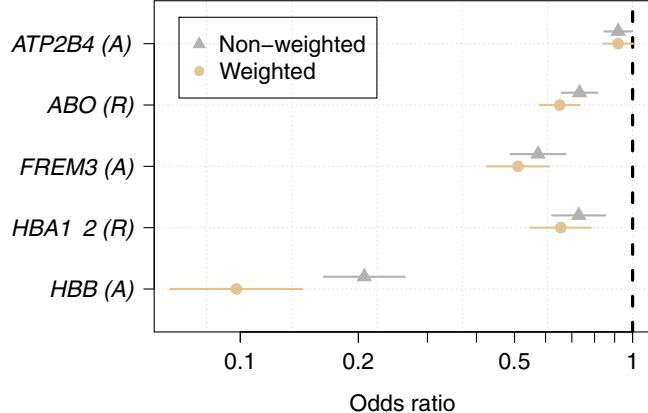

**Fig. 4 Non-weighted versus probabilistically weighted case-control analysis.** Odds-ratios (point estimates and 95% confidence intervals estimated under a logistic regression model) are shown for a case–control analysis for the five major red cell polymorphisms.

with slightly lower parasite densities, yet plasma concentrations of *Pf*HRP2 were higher, as was the sequestration index (the ratio of *Pf*HRP2 to parasite density). If this corresponds to a true effect, it could have several possible explanations. One would be that parasite multiplication rates are lower in *ATP2B4* homozygous patients, thus leading to higher plasma *Pf*HRP2 concentrations relative to circulating parasites. However, this explanation would imply a longer illness course, which in theory should be associated with a greater degree of anemia, but this was not observed in the

current study. Other potential explanations are that a greater amount of *Pf*HRP2 is released into the plasma from the red cell at schizogony because less *Pf*HRP2 is retained in the mutant cytoadherent ruptured ghost erythrocytes, or that the mutation results in enhanced *Pf*HRP2 production. These hypotheses are testable in laboratory experiments.

## Methods

**Study design and participants.** Individual written informed consent, including consent to the storage and re-use of samples and data, was provided by the parents of all study participants. Ethical approval for the study was granted by the Kenya Medical Research Institute / National Ethical Review Committee in Nairobi, Kenya (SCC1192) and the Oxford Tropical Ethical Review Committee in Oxford, UK (020-06). Re-use of existing, appropriately anonymized, human data does not require ethical approval under the Oxford Tropical Research Ethics Committee regulations (OxTREC).

The Kenyan severe malaria cohort has been described in detail previously[7]. Cases consisted of children aged <14 years who were admitted to the high dependency ward of Kilifi County Hospital between June 11th 1999 and June 12th 2008 with clinical features of severe falciparum malaria. Controls were infants aged 3–12 months who were born in the same study area as cases between 1st May 2006 and 30th April 2008 and were recruits to a cohort study investigating genetic susceptibility to a range of childhood diseases[27]. Controls were closely representative of cases in terms of sex, ethnic group, and geographic area of residence, but not age. The purpose of the controls was to provide background allele frequency estimates of the various genetic polymorphisms in the study population.

**Clinical and laboratory data.** Data and samples were collected through a clinical surveillance system that has been described in detail previously[28]. Severe malaria was defined as a positive blood-film for *P. falciparum* parasites together with prostration (a Blantyre Coma Score of 3 or 4), cerebral malaria (a Blantyre Coma Score of <3), respiratory distress (abnormally deep or "Kaussmaul's" breathing), severe anemia (hemoglobin concentration of <5 g/dL) or any other feature of severe

malaria as defined by the World Health Organization[29]. Admission blood films were stained and examined by standard methods[30] and parasite densities recorded as either the proportion of parasitized red blood cells in thin films (>1% of red cells parasitized) or based on the ratio of parasites to white blood cells in thick films for lower density infections. Parasite densities (the number of parasites/μl of whole blood) were then calculated with reference to full blood counts, if available, or if not, on the assumption of a white blood cell count of $8 \times 10^3$/μL or a red blood cell count of $5 \times 10^6$ /μL.

**Genotyping**. Severe malaria cases were genotyped for the rs334 SNP and for the common African form of α-thalassemia (a 3.7 kb deletion in *HBA1* and *HBA2*; -α$^{3.7}$-thalassemia[31]) by PCR at the KEMRI-Wellcome Trust Laboratories, Kenya, as described previously[7,32,33]. Genotyping for a further 119 candidate SNPs was conducted at the Wellcome Centre for Human Genetics in Oxford, UK, by use of the Sequenom iPLEX MassARRAY platform, using DNA extracted from frozen samples of whole blood as described previously[7]. Plasma concentrations of *Pf*HRP2 were batch-analyzed in Kenya by ELISA in admission samples stored at −80 °C using previously published methods[34]. We focused our analysis on 14 of the 15 previously associated genes: we excluded *INPP4B* because it tags the same nearby causal Dantu *GYP* mutation as *FREM3*, which is in stronger linkage disequilibrium[25,35].

**Statistical analysis**. We explored the effects of polymorphisms in *ABO* (rs8176719), *ATP2B4* (rs1541255), *G6PD* (rs1050828), *HBA1-2* (α−3.7-thalassemia), *HBB* (rs334), *FREM3/GYP* (rs186873296), *LPHN2* (rs72933304), *CAND1* (rs10459266), *RPS6KL1* (rs3742785), *GNAS* (rs8386), *ARL14* (rs75731597), *CD40LG* (rs3092945), *IL10* (rs1800890), and *LOC727982* (rs1371478) on three quantitative parasite-specific traits: (i) peripheral parasite density (the number of circulating parasites per μL); (ii) plasma *Pf*HRP2 concentration (ng/mL), a proxy measure of the total parasite biomass; and (iii) the ratio of the plasma *Pf*HRP2 concentration to the circulating parasite density (ng/parasite), which is proportional to the previously reported sequestration index[9]. Supplementary Table 1 provides genotype frequencies in cases and controls. The calculation of the sequestration index requires multiplying the plasma *Pf*HRP2 by a constant term which is an estimate of the multiplication rate over the previous life cycle[9]. The *Pf*HRP2 to parasite ratio is simpler to interpret as it does not make any assumptions about multiplication rates (which could potentially vary considerably across individuals), an issue that is particularly pertinent in the context of malaria-protective genes. For all three endpoints we used a $\log_{10}$ transformation as the parasite density and plasma *Pf*HRP2 concentration are both approximately log-normally distributed. The $\log_{10}$ *Pf*HRP2 to parasite ratio is thus expressed as the difference between the two $\log_{10}$ transformed variables. To aid the comparison of effect sizes across the three quantitative traits we standardized each dependent variable to obtain distributions with mean 0 and standard deviation 1. To account for parasite densities that were below the lower limit of detection by thick film microscopy (~50 parasites/μL), we used a $\log_{10}(x + 50)$ transformation for the peripheral parasite density. We excluded non-measurable *Pf*HRP2 concentrations (22/1522 samples) as these could represent *HRP2/3* parasite gene deletions or assay errors. Out of 2220 genotyped severe malaria patients, measurable *Pf*HRP2 concentrations were available for 1500 patients and parasite densities were available for 2184 patients. In addition, we included patient data on platelet counts, admission hemoglobin and other key admission variables relating to patient severity and demographics. To increase power, we performed multiple imputation using random forests for the patients with missing *Pf*HRP2 concentrations and missing parasite densities, excluding patients who had missing *Pf*HRP2 concentration, parasite count and platelet count data ($n = 22$ patients). Ten imputed datasets were generated (R package *missForest*). Two sensitivity analyses were performed: first, we included additional covariate adjustment for hemoglobin, age, and bacteremia; second, we estimated effect sizes for the same association models using only patients with non-missing phenotype data (complete case analysis).

Quantitative trait associations were assessed under linear models with Gaussian errors. All models were adjusted for self-reported ethnicity. We fitted a range of association models to the parasite density and plasma *Pf*HRP2-based traits including general two-parameter models, additive, heterozygous and recessive models, and determined the best fitting models from the model likelihoods. The latter three models of association were then compared against the general two-parameter model using a likelihood ratio test with 1 degree of freedom. A *p*-value threshold of 0.05 was used to reject an association model. All *p*-values are reported without correction for multiple testing. Parameter estimates for each imputed dataset were combined using Rubin's rules (R package *mitools*) and are given in Supplementary Tables 2–4. To demonstrate the phenotypic information contained in the plasma *Pf*HRP2 for the severe malaria cases, we estimated case-control odds-ratios for protection against severe malaria, using probabilistic data-tilting to down-weight patients who likely did not have severe malaria, using a previously reported model based on *Pf*HRP2 and platelet counts[13]. Tests for Hardy–Weinberg equilibrium were reported in our earlier study[7] for all 14 SNPs, with no significant departures from expected genotype frequencies. All analyses were done in R v4.0.2.

**Reporting summary**. Further information on research design is available in the Nature Research Reporting Summary linked to this article.

## Data availability

All data used in the main analyses are available at the following github repository: https://github.com/jwatowatson/Parasite_biomass. A permanent version has been archived on Zenodo: https://doi.org/10.5281/zenodo.6499851.

## Code availability

All code is available at the following github repository: https://github.com/jwatowatson/Parasite_biomass. A permanent version has been archived on Zenodo: https://doi.org/10.5281/zenodo.6499851.

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

## Acknowledgements

This research was funded, in whole or in part, by The Wellcome Trust, Grant 093956/Z/10/C. A CC-BY or equivalent license is applied to author accepted manuscript arising from this submission, in accordance with the grant's open access conditions. This work was conducted as part of SMAART (Severe Malaria Africa – A consortium for Research and Trials) funded by a Wellcome Collaborative Award in Science grant (209265/Z/17/Z) held in part by K.M. and A.M.D. Sample genotyping was conducted in part by the Malaria Genomic Epidemiology Network, supported by Wellcome (WT077383/Z/05/Z) and the Bill & Melinda Gates Foundation through the Foundations of the National Institutes of Health (566) as part of the Grand Challenges in Global Health Initiative. The Resource Centre for Genomic Epidemiology of Malaria is supported by Wellcome (090770/Z/09/Z; 204911/Z/16/Z). This research was supported by the Medical Research Council (G0600718; G0600230; MR/M006212/1). Wellcome also provides core awards to the Wellcome Centre for Human Genetics (203141/Z/16/Z) and the Wellcome Sanger Institute (206194). Sample collection and processing was further supported through a Programme Grant (092654) to the Kilifi Programme from Wellcome. T.N.W. and N.J.W. are senior and principal research fellows respectively, funded by the Wellcome Trust (202800/Z/16/Z and 093956/Z/10/C, respectively). J.A.W. is a Sir Henry Dale Fellow funded by the Wellcome Trust (223253/Z/21/Z). This paper is published with permission from the Director of the Kenya Medical Research Institute (KEMRI).

## Author contributions

S.U., J.A.W., N.J.W., and T.N.W. conceived the study. P.W., J.C.R., A.M.W., S.N.K., G.M.N., M.S., M.M., N.M., C.J.W., A.M.D., and K.W. curated data and provided study materials. J.A.W. analyzed the data. T.N.W. wrote the first draft of the manuscript. All authors approved the final manuscript.

## Competing interests

The authors declare no competing interests.
