## [Peer Review File · Nature Communications]

Peer review comments first round -

Reviewer #1 (Remarks to the Author):

Severe falciparum malaria is a major cause of preventable child mortality in sub-Saharan Africa. The sequestration of parasitized erythrocytes in the microvasculature of vital organs is a central pathophysiological feature. The plasma concentration of the parasite protein P. falciparum Histidine-Rich Protein 2 (PfHRP2) has diagnostic and prognostic value in severe malaria. In the current study we investigate the potential use of plasma PfHRP2 and the sequestration index (the ratio of plasma PfHRP2 to circulating parasites) as quantitative traits in the conduct of case-only genetic association studies of severe malaria. The authors demonstrate the utility of this approach using data from over 2,000 Kenyan children with severe malaria, genotyped for 14 major candidate genes that were found to be associated with protection against severe malaria in previous studies.

The manuscript can benefit from the following comments

Abstract

1. The authors can include some stats to support the results presented, more so, the most significant findings.
2. Include a conclusion statement based on the results presented.

Methods

3. The control group needs to be defined more. Who formed the control group? How were they recruited? Were they age- and gender-matched? Why did we match 3-12 months children against <14 years? What age do the children acquire clinical immunity in this population? Was this a consideration in the matching of cases and controls?
4. When doing parasite densities recorded as either the ratio of parasites to white blood cells in thick films or the proportion of parasitised red blood cells in thin films for heavier infections, did we get different parasite densities using these 2 methods? What did we do with the 2 different parasite densities for the same individual?
5. I have a feeling that there were several factors that could have been controlled for in the current association analyses eg HIV status, co-infections with other pathogens like bacteremia, gender, age, presence of intestinal worms etc. I checked out for the exclusion criteria and I couldn't pick put any.
6. Given that there were a huge disparity between the ages of children in cases and controls, an age-dependent/guided analyses should have been performed.
7. Based on the current findings, what could potentially be hypothesized (in the discussion) on children with mixed clinical presentations.

Reviewer #2 (Remarks to the Author):

The study by Uyoga et al. presents some very interesting findings about the use of plasma PfHRP2 as quantitative trait in genetic association studies of severe malaria. The study included a large dataset from over 2,000 Kenyan children with severe malaria. The manuscript is concise and generally very well-written, but I have some important concerns with regard to the methods used and results/discussion presented.

Major comments

1. Provide a table with the minor allele frequency (MAF) for each SNP, genotype distributions and HWE estimates. HWE deviations can signal important problems, as genotyping errors, but also can have other explanations as population stratification and selection bias.

2. This reviewer is concerned about possible biases that imputed dataset might have led to the association models. Hong and Lynn pointed out this issue in their article recently published (2020, BMC Medical Research Methodology, <https://doi.org/10.1186/s12874-020-01080-1>), in which they demonstrated that missForest can produce severely biased regression coefficient estimates. Have the association analyses been conducted with non-imputed dataset and compared to the analyses conducted with imputed dataset? At this time, it is not clear whether some differences that have been reported in the present manuscript in relation to results obtained by others were due to inappropriate methodological approach. From methods description, we can see that imputed data for plasma PfHRP2 concentration were generated for 698 individuals of 2198 subjects (~32%) included in the association analysis. This issue needs to be appropriately addressed by authors.

3. Concerning the relationships found for ATP2B4 (rs1541255), it does not seem to be an unexpected finding. Ndiaye et al (<https://www.medrxiv.org/content/10.1101/2021.12.03.21267245v1.full#ref-31>) found very interesting results involving this and four other SNPs (rs11240734, rs1541252, rs1541253, rs1541254, and rs1541255), which exhibits a strong LD pattern among them. They found some haplotypes are associated with an increased risk of developing severe malaria and that these SNPs are placed in a region of 506 bp enough to regulate the expression of ATP2B4. These findings may support your results regarding higher plasma concentrations of PfHRP2, as well as the sequestration index, associated to the polymorphism in ATP2B4.

4. It is important to include consideration on the statistical power of the GWAS sample size: power calculation prior to a genetic experiment determine the optimal sample size to detect a disease-associated locus. This information should be inserted in the Methods.

5. Please, add a supplementary table with the regression coefficients and their p-values for the best fitting models.

Minor comments

6. In Genotyping section, includes the “rs” of each SNP analyzed in the present study. Currently, we can found a description of them only in the figure captions.

7. Caption of Figure 1 – needs to be better explained. Its definition is clear only from the body of the manuscript.

8. Captions of Figure 2 and 3 – Indicate what is represented in the plots (median and quartiles?), scale of y-axis (log10?), the number of individuals in each group and any statistical difference between groups.

Reviewer #3 (Remarks to the Author):

Nature Communication: Review Report

Title: The impact of malaria-protective red blood cell polymorphisms on parasite biomass in children with severe *Plasmodium falciparum*

ID: Nature-com355180_0_art_0_r809hy

General comments

The authors set out to investigate candidate genes, mostly related to red blood cells' structure or function, that provide protection against severe *P. falciparum* malaria and correlate with the peripheral parasite density. The investigation suggests that plasma PfHRP2 and the ratio of the plasma PfHRP2 to the peripheral parasite density, (sequestration index) as both diagnostic and prognostic of severe *P. falciparum* malaria disease. While the relationship between pfHRP2 and severe malaria is known, the proposition on the sequestration index is novel and very interesting. Since peripheral parasitemia need to be assessed microscopically, the authors could indicate the limitation in its utility, if assessment of periphery parasitemia is inefficiently done. Would it make a difference if PCR or qPCR is used to assess the peripheral parasitemia instead of microscopy? Additionally, the authors did not mention how the PfHRP2 was measured, although the reference for Dondorp AM, 2005 indicates the use of ELISA.

Specific comments

An important aspect of the current proposal by the authors is the Pf parasite biomass. Recently, there was an article that indicated that the spleen harbors a significant proportion of the parasite biomass (Sho S, 2021). This was highest for Pv and then Pf. Interestingly, the entire asexual life cycles were observed to occur in the spleen, which means the PfHRP2 could indicate perhaps the role of the spleen in malaria disease severity and how it varies between individuals and populations.

Recommendation

The article provides additional information on how parasite biomass could be assessed in *Plasmodium* infections. This is a very important contribution that could assist with detection of the profile of individuals who may be at risk of severe Pf malaria.

Isaac Q.

Reviewer #1 (Remarks to the Author)

Severe falciparum malaria is a major cause of preventable child mortality in sub-Saharan Africa. The sequestration of parasitized erythrocytes in the microvasculature of vital organs is a central pathophysiological feature. The plasma concentration of the parasite protein P. falciparum Histidine-Rich Protein 2 (PfHRP2) has diagnostic and prognostic value in severe malaria. In the current study we investigate the potential use of plasma PfHRP2 and the sequestration index (the ratio of plasma PfHRP2 to circulating parasites) as quantitative traits in the conduct of case-only genetic association studies of severe malaria. The authors demonstrate the utility of this approach using data from over 2,000 Kenyan children with severe malaria, genotyped for 14 major candidate genes that were found to be associated with protection against severe malaria in previous studies.

The manuscript can benefit from the following comments

Abstract

1. The authors can include some stats to support the results presented, more so, the most significant findings.
2. Include a conclusion statement based on the results presented.

We thank the reviewer for their comments. Unfortunately, Nature Communications requires that the abstract is at most 150 words. Our original submission had 250 words, so we have had to cut rather than add to the abstract.

The online author guide states: *“The abstract — which should be no more than 150 words long and contain no references — should serve both as a general introduction to the topic and as a brief, non-technical summary of the main results and their implications.”*

Methods

3. The control group needs to be defined more. Who formed the control group? How were they recruited? Were they age- and gender-matched? Why did we match 3-12 months children against <14 years? What age do the children acquire clinical immunity in this population? Was this a consideration in the matching of cases and controls?

We agree that the question of age matching is important. However, whether or not controls should be age-matched strongly depends on the desired interpretation of the case-control odds-ratio: the standard approach in severe malaria has been to select controls who represent the background population genotype frequencies *at birth* (i.e. before any selection pressure from malaria which predominantly kills young children in high transmission areas). For a particular allele, unless there are indirect effects of the mother's genotype in pregnancy and protective effects of the child's genotype within the womb, we would expect Hardy-Weinberg equilibrium predicted genotype frequencies at birth. If we sampled older children, the genotype frequencies would change as a function of the protective effects. Thus, sampling controls using, for example, cord blood allows us to estimate effect sizes relative to approximate Hardy-Weinberg genotypes frequencies. We note that many previously published severe malaria case-control studies have used the same approach (controls are either cord blood or young infants), including previous studies by the MalariaGen Consortium (eg Nature Genetics 2014, PMID: PMC4617542).

We note also that in this context, controls are not “not cases” but represent the background expected genotype frequencies (i.e. some of the controls could develop severe malaria or have already have had severe malaria for the older infants). This is different from standard case-control studies in non-communicable diseases where controls are considered as “not cases”.

We have added (lines 211-216):

“Controls were infants aged 3–12 months who were born in the same study area as cases between 1st May 2006 and 30th April 2008 and were recruits to a cohort study investigating genetic susceptibility to a range of childhood diseases [REF]. Controls were closely representative of cases in terms of sex, ethnic group, and geographic area of residence, but not with regard to age. The purpose of the controls was to provide background allele frequency estimates of the various genetic polymorphisms in the study population.”

We note that our main results concern case-only genetic associations using quantitative phenotypic traits for the cases (parasite density, PfHRP2 and the PfHRP2/parasitaemia ratio). For these analyses, the question of age-matching does not apply as data from controls are not used. Controls are only used for the final analysis (Figure 4) which illustrates how mis-diagnosis of severe malaria (specifically those patients with low PfHRP2) can influence the results of case-control genetic association analysis (bias towards the null).

4. When doing parasite densities recorded as either the ratio of parasites to white blood cells in thick films or the proportion of parasitised red blood cells in thin films for heavier infections, did we get different parasite densities using these 2 methods? What did we do with the 2 different parasite densities for the same individual?

Thank you for this question which highlights a lack of clarity on our part about a method that is standard in malaria parasitology. The methods used were mutually exclusive from the point of view of parasite density reporting. Our SOP with regard to parasite counting means that heavy infections are reported from counts of thin blood films while light infections are reported from counts of thick blood films. “Heavy” is defined as >1% of red blood cells infected, as judged by the examination of 20 microscopy fields. Neither method is perfect and both involve variability based on the quality of the blood films and the area of films that are examined, but in the context of epidemiological studies they give broadly consistent results. We have modified the methods (lines 229-232) to read as follows:

“Admission blood films were stained and examined by standard methods¹⁷ and parasite densities recorded as either the proportion of parasitised red blood cells in thin films (>1% of red cells parasitized) or based on the ratio of parasites to white blood cells in thick films for lower density infections.”

5. I have a feeling that there were several factors that could have been controlled for in the current association analyses eg HIV status, co-infections with other pathogens like bacteremia, gender, age, presence of intestinal worms etc. I checked out for the exclusion criteria and I couldn't pick put any.

HIV prevalence among children in this population is very low (<1%). All the factors mentioned could be predictors of plasma PfHRP2 because they predict whether a patient truly has severe malaria or not. However, we note that they are only confounders if they were correlated with population stratification in cases (which is possible). Bacteraemia is recorded but blood culture has very low sensitivity.

We did a sensitivity analysis with additional adjustment for age, sex, haemoglobin, and bacteraemia. The results are near identical to those of the main analysis and are shown in the Supplementary Figure 2.

6. Given that there were a huge disparity between the ages of children in cases and controls, an age-dependent/guided analyses should have been performed.

See our reply above for why we do not think that age-stratification is necessary or appropriate.

7. Based on the current findings, what could potentially be hypothesized (in the discussion) on children with mixed clinical presentations.

Our apologies to the reviewer but we don't fully understand the point that is being made. Like children with any severe clinical illness, the clinical features seen among children with severe malaria are often diverse. There is considerable aetiological overlap with regard to the most common clinical features of severe malaria. For example, critically ill children with severe metabolic acidosis due to severe anaemia can manifest both deep breathing (a consequence of the acidosis) and prostration / coma (a late-stage feature of many severe illnesses). We have no specific hypotheses to put forward in the context of this analysis.

Reviewer #2 (Remarks to the Author):

The study by Uyoga et al. presents some very interesting findings about the use of plasma PfHRP2 as quantitative trait in genetic association studies of severe malaria. The study included a large dataset from over 2,000 Kenyan children with severe malaria. The manuscript is concise and generally very well-written, but I have some important concerns with regard to the methods used and results/discussion presented.

Major comments

1. Provide a table with the minor allele frequency (MAF) for each SNP, genotype distributions and HWE estimates. HWE deviations can signal important problems, as genotyping errors, but also can have other explanations as population stratification and selection bias.

We thank the reviewer for this suggestion. We have added Table 1 which shows the genotype frequencies for the 14 targeted polymorphisms in the controls and the cases. We have not added Hardy-Weinberg tests for the following reasons: for a polymorphism that truly protects against severe malaria (e.g. heterozygous haemoglobin S), we expect departures from the Hardy-Weinberg equilibrium (HWE) frequencies due to selection (e.g. heterozygous haemoglobin S patients are much rarer than expected from population allele frequencies). Indeed, testing for HWE in cases is an indirect test for selection which can in some contexts be quite powerful. In our previous study (Ndila et al, Lancet Haematology) we checked all polymorphisms for departures from HWE in the controls (where, as the reviewer points out, departures from HWE could suggest genotyping errors or population stratification). We have added to the Methods (lines 297-298):

“Tests for Hardy-Weinberg equilibrium were previously done in our earlier study⁷ for all 14 SNPs, with no significant departures from expected genotype frequencies.”

2. This reviewer is concerned about possible biases that imputed dataset might have led to the association models. Hong and Lynn pointed out this issue in their article recently published (2020, BMC Medical Research Methodology, <https://doi.org/10.1186/s12874-020-01080-1>), in which they demonstrated that missForest can produce severely biased regression coefficient estimates. Have the association analyses been conducted with non-imputed dataset and compared to the analyses conducted with imputed dataset? At this time, it is not clear whether some differences that have been reported in the present manuscript in relation to results obtained by others were due to inappropriate methodological approach. From methods description, we can see that imputed data for plasma PfHRP2 concentration were generated for 698 individuals of 2198 subjects (~32%) included in the association analysis. This issue needs to be appropriately addressed by authors.

This is an important point. We have added a complete-case analysis (the results of which are shown in Supplementary Figures 3 & 4). The complete case analysis shows near identical results with only slight differences in the confidence intervals (tighter for the multiple imputation due to the additional information used). We hope that this will alleviate the

reviewer's concerns regarding missForest.

3. Concerning the relationships found for ATP2B4 (rs1541255), it does not seem to be an unexpected finding. Ndiaye et al (<https://www.medrxiv.org/content/10.1101/2021.12.03.21267245v1.full#ref-31>) found very interesting results involving this and four other SNPs (rs11240734, rs1541252, rs1541253, rs1541254, and rs1541255), which exhibits a strong LD pattern among them. They found some haplotypes are associated with an increased risk of developing severe malaria and that these SNPs are placed in a region of 506 bp enough to regulate the expression of ATP2B4. These findings may support your results regarding higher plasma concentrations of PfHRP2, as well as the sequestration index, associated to the polymorphism in ATP2B4.

We regret that we did not understand this reviewer comment. Ref 31 in the Nisar et al paper linked by the reviewer is our own previous study published in Lancet Haematology (Ndila et al). We could not see a reference in the linked paper for which Ndiaye was first author. It is unclear to us how the results from Nisar et al could be linked to lower plasma PfHRP2 concentrations in patients with severe malaria.

4. It is important to include consideration on the statistical power of the GWAS sample size: power calculation prior to a genetic experiment determine the optimal sample size to detect a disease-associated locus. This information should be inserted in the Methods.

We agree that power analyses are important. However, the utility of a power analysis in this context is unclear for the following reasons:

1. This is a retrospective analysis of existing data, with no experiments planned
2. A power calculation requires three assumptions:
 - a. The effect size (as a fold change in PfHRP2 for example)
 - b. The genetic model (additive, recessive or heterozygote)
 - c. The allele frequency

Regarding the second point, the scenarios which are present in the data vary considerably. For *HBB* we have an additive model with a very large effect size and an allele frequency of ~8.5%. For the O blood group and alpha-thalassaemia, the allele frequencies are much higher (~40%), but the effect sizes are much smaller and the best fitting models are recessive. Together, such issues make it impossible to develop a sensible power-size calculation a priori.

5. Please, add a supplementary table with the regression coefficients and their p-values for the best fitting models.

We have added these to the supplementary materials, Tables 1-3.

Minor comments

6. In Genotyping section, includes the "rs" of each SNP analyzed in the present study. Currently, we can find a description of them only in the figure captions.

We agree that this would be a significant improvement and have added these to the Methods (see lines 249-252) and to Table 1:

“We explored the effects of polymorphisms in ABO (rs8176719), ATP2B4 (rs1541255), G6PD (rs1050828), HBA1-2 (α -³⁻⁷-thalassaemia), HBB (rs334), FREM3/GYP (rs186873296), LPHN2 (rs72933304), CAND1 (rs10459266), RPS6KL1 (rs3742785), GNAS (rs8386), ARL14 (rs75731597), CD40LG (rs3092945), IL10 (rs1800890) and LOC727982 (rs1371478) on three quantitative parasite-specific traits: (i) peripheral parasite density (the number of circulating parasites per μ L); (ii) plasma PfHRP2 concentration (ng/mL), a proxy measure of the total parasite biomass; and (iii) the ratio of the plasma PfHRP2 concentration to the circulating parasite density (ng/parasite), which is proportional to the previously reported sequestration index⁹.”

7. Caption of Figure 1 – needs to be better explained. Its definition is clear only from the body of the manuscript.

The caption of Figure 1 has been changed. It now reads:

“Case-only genetic association study using 14 targeted variants and three quantitative traits. a-c Distributions of the three quantitative traits (observed data). d Effect estimates (95% confidence intervals) expressed as fold changes on the standardised trait distribution (rescaled to have mean 0 and standard deviation 1).”

8. Captions of Figure 2 and 3 – Indicate what is represented in the plots (median and quartiles?), scale of y-axis (log10?), the number of individuals in each group and any statistical difference between groups.

We have added an explanation of the boxplots (median, interquartile range, and whiskers), and we have added the number of patients per group. We did not add significance tests across groups as this makes the plot very busy. In addition, associations are tested across the three genotypes under a linear model assuming additive, heterozygote or recessive effects, which is different from simple pairwise comparisons.

Reviewer #3 (Remarks to the Author):

General comments

The authors set out to investigate candidate genes, mostly related to red blood cells' structure or function, that provide protection against severe *P. falciparum* malaria and correlate with the peripheral parasite density. The investigation suggests that plasma PfHRP2 and the ratio of the plasma PfHRP2 to the peripheral parasite density, (sequestration index) as both diagnostic and prognostic of severe *P. falciparum* malaria disease. While the relationship between pfHRP2 and severe malaria is known, the proposition on the sequestration index is novel and very interesting. Since peripheral parasitemia need to be assessed microscopically, the authors could indicate the limitation in its utility, if assessment of periphery parasitemia is inefficiently done. Would it make a difference if PCR or qPCR is used to assess the peripheral parasitemia instead of microscopy? Additionally, the authors did not mention how the PfHRP2 was measured, although the reference for Dondorp AM, 2005 indicates the use of ELISA.

We thank the reviewer for their kind comments. PCR could be used to assess parasite densities but quantitative results are not reliable for the very high parasite densities seen in severe malaria, so microscopy is preferred. Another problem with PCR is that it requires calibration using fixed volumes of blood. The samples available to us for the assessment of plasma PfHRP2 were frozen samples of packed cells which does not allow for accurate assessment of parasite densities. All patients with severe malaria should have circulating parasites densities above the lower limit of detection of microscopy.

Regarding the plasma PfHRP2 we state in the Methods (line 242-243):

"Plasma concentrations of PfHRP2 were batch-analysed in Kenya by ELISA in admission samples stored at -80°C using previously published methods²⁰."

Specific comments

An important aspect of the current proposal by the authors is the Pf parasite biomass. Recently, there was an article that indicated that the spleen harbors a significant proportion of the parasite biomass (Sho S, 2021). This was highest for Pv and then Pf. Interestingly, the entire asexual life cycles were observed to occur in the spleen, which means the PfHRP2 could indicate perhaps the role of the spleen in malaria disease severity and how it varies between individuals and populations.

We believe the reviewer is referring to Kho et al (NEJM, 2021). This is indeed an important observation, however plasma PfHRP2 concentrations provide no indication as to where the parasites originate from. Conversion from plasma PfHRP2 levels to parasites per unit of blood requires making assumptions about the multiplication rate during the previous cycle and would assume that parasites sequestered in the spleen behave in the same way as the other parasites.

Recommendation

The article provides additional information on how parasite biomass could be assessed in Plasmodium infections. This is a very important contribution that could assist with detection

of the profile of individuals who may be at risk of severe Pf malaria.
Isaac Q.

Peer review comments second round -

Reviewer #1 (Remarks to the Author):

None

Reviewer #2 (Remarks to the Author):

After the authors have satisfactorily addressed most of the concerns raised by reviewers, I recommend the publication of manuscript by Uyoga et al. I also congratulate the authors for their important findings.

Reviewer #3 (Remarks to the Author):

I think that the authors have adequately responded to the comments made by the reviewers in the revised manuscript.